# Atomic Oxygen Adaptability of Flexible Kapton/Al$_2$O$_3$ Composite Thin Films Prepared by Ion Exchange Method

**Donghua Jiang** [1], **Dan Wang** [1,2], **Gang Liu** [3] **and Qiang Wei** [1,4,*]

[1] Tianjin Key Laboratory of Composite and Functional Materials, School of Materials Science and Engineering, Tianjin University, Tianjin 300350, China; jdhua_tz@tju.edu.cn (D.J.); wangdan@tju.edu.cn (D.W.)

[2] Jizhong Energy Resources Co., LTD, Xingtai 054000, China

[3] Department of environment, Shanghai Institute of Satellite Equipment, Shanghai 200240, China; elest@126.com

[4] School of Mechanical Engineering, Hebei University of Technology, Tianjin 300130, China

[*] Correspondence: weiqiang@hebut.edu.cn

**Abstract:** Polyimide film (Kapton) is an important polymer material used for the construction of spacecrafts. The performance of Kapton can be degraded for atomic oxygen erosion in space. Commonly used atomic oxygen protective layers have issues such as poor toughness and poor adhesion with the film. In this paper, Kapton/Al$_2$O$_3$ nanocomposite films were prepared via an ion exchange method, and the optical properties, mechanical properties, and mechanisms for the change in the mass and microstructure, before and after atomic oxygen exposure, were analyzed. The results show that the deposition of the Kapton/Al$_2$O$_3$ surface nanocomposite film prepared via the ion exchange method has no obvious effects on the internal structure and optical transmittance of the Kapton film matrix. The tensile strength and elongation of the prepared film were much higher than those of the pure Kapton film, demonstrating its good flexibility. Scanning electron microscope (SEM) analysis showed that the etching pits had a carpet-like morphology on the composite film surface and were relatively small after atomic oxygen erosion. In contrast with the C–C bond rupture in the oxydianiline (ODA) benzene in Kapton films, the Kapton/Al$_2$O$_3$ nanocomposite film mainly destroyed the C=C bond in the pyromellitic dianhydride (PMDA) benzene ring. On exposure to an atomic oxygen environment for a short period, the Kapton/Al$_2$O$_3$ nanocomposite film exhibited improved atomic oxygen erosion resistance because the Al$_2$O$_3$ layer inhibited atomic oxygen diffusion. With increasing atomic oxygen exposure time, the atomic oxygen diffused into the Kapton matrix via the pores of the Al$_2$O$_3$ layer, causing damage to the substrate. This resulted in a detachment of the surface Al$_2$O$_3$ layer and exposure of the Kapton matrix, and thereby the atomic oxygen resistance was decreased. The applicability of the ion exchange mechanism of trivalent Al element on the surface modification of the polyimide is explored in this study. The behavior of the Kapton/Al$_2$O$_3$ composite film under the atomic oxygen environment of space is investigated, which provides the basis for studying the effects of atomic oxygen on the flexible protective Kapton film.

**Keywords:** atomic oxygen adaptability; Kapton; flexible; ion exchange method

## 1. Introduction

Polyimide films have been widely used for the development and application of components including thermal control coatings, flexible solar panels, and thin film mirrors, owing to their excellent resistance to high and low temperature, radiation resistance, chemical stability, insulation, and high strength [1,2]. Atomic oxygen in the low earth orbit environment is the major reason for the degradation

of the properties of the spacecraft material. Under the action of atomic oxygen, the polyimide film suffers a degradation in its performance, such as mass loss, surface roughening, and a reduction in strength, which seriously affects the operation of the spacecraft [3,4]. Preparation of protective layers, such as silicon oxide and aluminum oxide on the surface of polyimide have been carried out via different methods such as in-situ deposition, sol-gel, magnetron sputtering, and ion plating [5–8] to improve antigenic oxygen erosion [9–11]. However, the poor toughness and poor adhesion properties of these protective coatings make them prone to defects, such as cracks and peeling. Therefore, the polyimide substrate is susceptible to stress corrosion cracking. The introduction of an organic group to prepare an organic/inorganic composite coating can solve the above issues. However, the thickness of the composite coating decreases after the organic coating reacts with atomic oxygen, fine cracks form and the strength is lowered. Additionally, the organic coating is also affected by ultraviolet radiation and vacuum, leading to its deterioration [12–19].

Chemical surface modification of the polyimide via the ion exchange method can form a metal or metal oxide functional nanocomposite layer on the surface of the film, without changing its original properties, such as the polyimide's flexibility [20,21]. This method comprises the surface hydrolysis of commercial polyimide (PI) film in an alkali solution, resulting in the cleavage of the imide rings and the formation of carboxylic acid groups, subsequent loading of metal ions through the ion exchange of the carboxyl group with the metal cations in the inorganic metal salt aqueous solution, and final thermal treatment of the hybrid film in ambient atmosphere to generate the surface metallized PI nanocomposite films [22].

The literature shows that for low-valent metal elements with monovalent or divalent metal salt ions, nanocomposite films such as PI/Ag, PI/NiO, PI/Cu, PI/ZnO, and PI/$Co_3O_4$ have been successfully prepared via the ion exchange method [22–29], which make full use of the functional properties of the metal oxide nanoparticles, while maintaining the excellent mechanical and thermal properties of PI. However, there are only a few reports on the applicability of higher-valent metal elements, such as trivalent Al ion. Alumina ($Al_2O_3$) is a material with strong insulation, high thermal conductivity and strong oxidation resistance [30]. $Al_2O_3$ particles embedded in polyimide film will help to improve the space atomic oxygen adaptability of PI. In this work, to maintain the flexibility of the polyimide film and simultaneously improve its compatibility with the atomic oxygen environment in space, a Kapton/$Al_2O_3$ surface nanocomposite film was prepared via the ion exchange method. The suitability of the ion exchange method for amine surface modification with trivalent Al was explored. In addition, the practical behavior of the Kapton/$Al_2O_3$ composite films in an atomic oxygen environment mimicking space was evaluated.

## 2. Experiment

### 2.1. Experimental Materials

The matrix material was a polyimide film (Kapton) commercially obtained from DuPont of the United States (Midland, MI) with a thickness of about 50 μm. It was synthesized from a dibasic anhydride and a diamine (PMDA-ODA). The molecular formula of the Kapton is shown in Figure 1. The Kapton film was cut into small pieces with a size of 10 mm ×10 mm; it was then ultrasonically cleaned with acetone and absolute ethanol for 10 min to remove residual organic pollutants on the surface of the material. After washing, it was dried naturally in air. The Kapton film was formed via the tap casting method, and the optical properties and roughness of the two sides of the film were different. For this work, the side that was relatively smooth and reflective was selected to be the test surface.

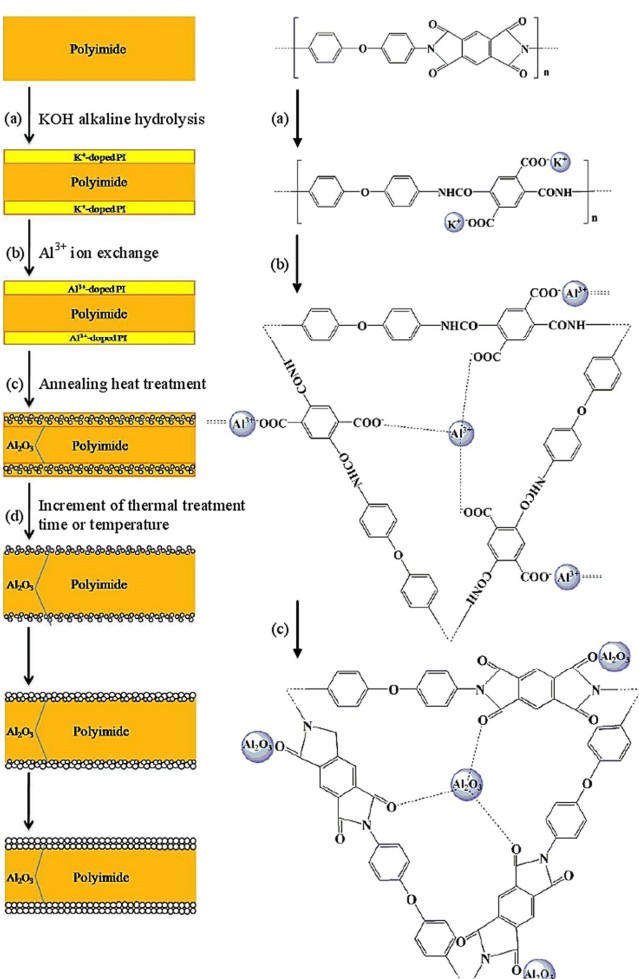

**Figure 1.** Schematic diagram of the molecular structure of the polyimide.

## 2.2. Preparation of Kapton/Al₂O₃ Composite Film

The Kapton film sample was immersed in a 5 M potassium hydroxide solution at room temperature for the hydrolysis reaction. After 5 min, the sample was taken out and thoroughly washed with a large amount of deionized water to remove the residual lye on the surface. Then, the hydrolyzed Kapton film was immersed in a 0.5 M aqueous solution of AlCl₃; it was taken out after 120 min and thoroughly washed with a large amount of deionized water to remove any residual metal salt solution on the surface; then, it was naturally dried in air. Subsequently, the aluminum ion doped Kapton film was subjected to an oxygen-containing heat treatment in a high-temperature controllable tube furnace (GSL-1300X, Kejing Equipment Manufacturing Co., Ltd, Shenyang, China). It was first heated from room temperature to 135 °C for 1 h to remove moisture in the material, and then heated up to 340 °C at a heating rate of 2 °C/min. After 2 h of soaking, the sample was cooled to room temperature in the furnace. Finally, a Kapton/Al₂O₃ nanocomposite film was formed on the Kapton substrate. The principle of preparing the Kapton/Al₂O₃ composite film is shown in Figure 2.

**Figure 2.** Schematic diagram of the reaction process for the Kapton/Al₂O₃ nanocomposite film prepared via the surface-modification-ion-exchange method.

### 2.3. Atomic Oxygen Exposure Test

The atomic oxygen exposure test was carried out using atomic oxygen from a microwave plasma source (AO-1, Flying Sail Plasma Technology Co., Ltd., Hefei, China). The device uses microwave discharge to ionize oxygen and form oxygen plasma that attacks a metal target with a negative voltage under the action of a magnetic field; in this way, the oxygen plasma is neutralized and reduced to obtain a neutral atomic oxygen beam. The average energy is ~5 eV and the flux was adjusted to be approximately $1 \times 10^{16}$ atoms/cm$^2$·s (error ± 10%). The Kapton film samples were subjected to atomic oxygen exposure tests for 0, 6, 12, 18, 24, and 30 h, respectively.

### 2.4. Testing and Characterization

The mechanical properties of the film were evaluated by an INSTRON 8784 tester (Norwood, MA, USA). The tensile properties before and after film treatment were measured according to GB/T 1040.3-2006 [31]. The tensile speed was 5 mm/min and the tensile test specimen was a dumbbell shape, as shown in Figure 3.

A UV/Vis dual-beam spectrophotometer (Model 4802, UNICO, Shanghai, China) was used to measure the optical transmittance of the film. The test wavelength ranged from 200 to 1100 nm with a resolution of 0.5 nm. The surface morphology of the film was observed via field emission scanning electron microscopy (SEM, Model S4800, HITACHI, Tokyo, Japan) equipped with an energy dispersive spectrometer (EDS), and it had an acceleration voltage of 10 kV. The sample surface was gold coated (palladium-gold alloy with a particle size of ~20 nm) before observation. The structure of the film surface was identified via Fourier transform infrared spectrometer (Nicolet Is10, Thermo Fisher Scientific, Waltham, MA, USA). A Ge crystal was used and the attenuated total reflectance-Fourier transform infrared (ATR-FTIR) spectrum ranged from 2000 to 400 cm$^{-1}$. X-ray photoelectron spectroscopy (Thermo ESCALAB 250XI, PHL, Waltham, MA, USA) was used to analyze the surface elemental composition and chemical valence of the film. The X-ray source was Mg K$\alpha$ with a power of 250 W and a grazing angle of 30°.

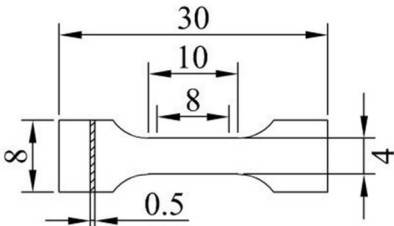

**Figure 3.** Dimensions of tensile specimens (unit: mm).

For the mass loss analysis, the samples were washed with absolute ethanol and dried before atomic oxygen exposure, and placed under a vacuum for more than 24 h. To prevent water influencing the absorption of the sample, the mass test before and after the atomic oxygen exposure was performed immediately after the sample was taken out of the vacuum chamber. The mass of the sample was measured using an electronic analytical balance (METTLER X-56 with an accuracy of 0.1 mg, Columbus, OH, USA), and the density of the sample was measured with a density measuring module (Density kit, ME-DNY-4, METTLER). The atomic oxygen reaction coefficient ($R_e$) of the sample was calculated according to the following formula [32]:

$$R_e = \frac{\Delta m}{\varphi} \cdot \frac{1}{\rho \cdot A} \tag{1}$$

where $\Delta m$ is the mass change of the sample before and after oxygen exposure, g; $\varphi$-atomic oxygen dose, atoms/cm$^2$; $\rho$-sample density, 1.5 g/cm$^2$; $A$-exposure area of the sample, 1 cm$^2$.

## 3. Kapton/Al$_2$O$_3$ Nanocomposite Film

### 3.1. Macroscopic Observation and Optical Performance Characterization

From Figure 4a, it can be seen that the Kapton film is transparent with a light brownish yellow color, and the test surface is glossy. After the ion exchange treatment, the as-prepared Kapton/ Al$_2$O$_3$ nanocomposite film turned dark brown (as shown in Figure 4b), but still maintained a certain transparency and glossiness [33,34]. The optical transmittance of the Kapton film and the Kapton/Al$_2$O$_3$ nanocomposite film in the wavelength range of 400–800 nm is shown in Figure 5. The cutoff wavelength of the Kapton film is ~500 nm. The transmittance is 0 over the wavelength range of 400–500 nm, while there is still some level of transmittance over the wavelength range of 500 to 800 nm and it exhibits a parabolic increasing relationship with increasing wavelength. In contrast, the overall trend for the transmittance of the Kapton/Al$_2$O$_3$ nanocomposite film remains basically unchanged; over the wavelength range of 500–800 nm, the transmittance decreases by ~10%. This may be attributed to the enhancement of diffuse reflection caused by the presence of Al$_2$O$_3$ particles on the surface.

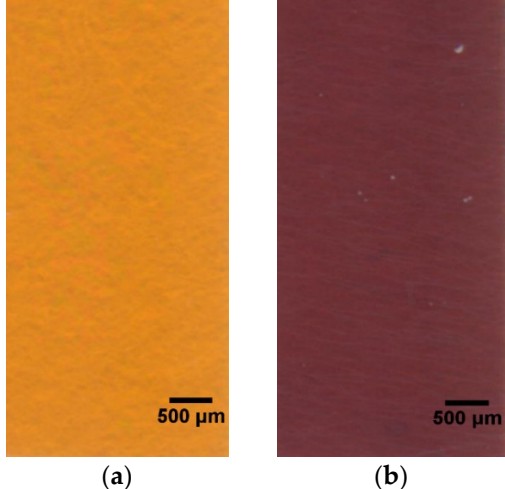

(**a**)　　　　　　　　　　　　(**b**)

**Figure 4.** Macroscopic picture of Kapton film (**a**) and Kapton/Al$_2$O$_3$ nanocomposite film (**b**).

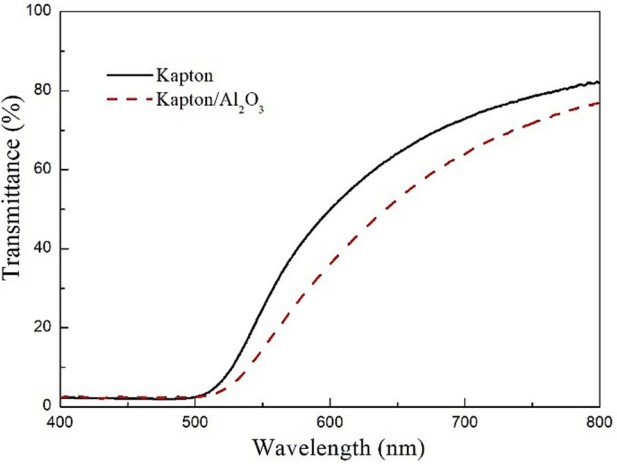

**Figure 5.** Optical transmittance of the Kapton film and Kapton/Al$_2$O$_3$ surface nanocomposite film.

### 3.2. Characterization of Mechanical Properties

Figure 6 shows the stress-strain curves of the Kapton film and the Kapton/Al$_2$O$_3$ composite film. It can be seen that the tensile strength and elongation of the Kapton/Al$_2$O$_3$ composite film are much higher than those of the pure Kapton film. To further study the toughness of Kapton/Al$_2$O$_3$ surface

nanocomposite film, the composite film was wound on a round bar with a diameter of 2 mm for 30 s, and then spread. The surface state of the film was observed and were compared with the picture before curling. The results are shown in Figure 7. The surface morphology of the Kapton/Al$_2$O$_3$ composite film before and after crimping was partially scratched. The surface of the composite film after crimping did not show cracks and peeling defects. The modified Kapton/Al$_2$O$_3$ surface nanocomposite film showed good toughness. The high strength and toughness of the nanocomposite films could be attributed to the initiation and propagation of cracks on the surface effectively prevented by Al$_2$O$_3$ particles in the subsurface layer for existence of phase interface, thus improving the overall strength and toughness of the composite films.

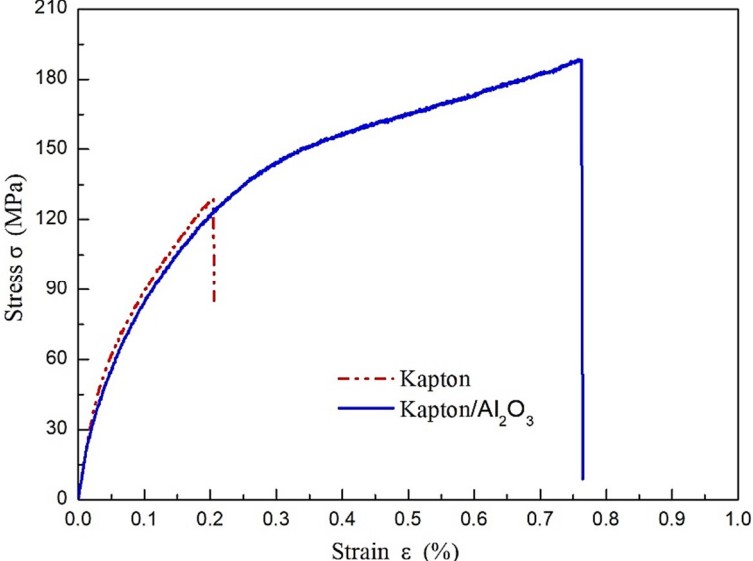

**Figure 6.** Stress–strain curve of Kapton film and Kapton/Al$_2$O$_3$ composite film.

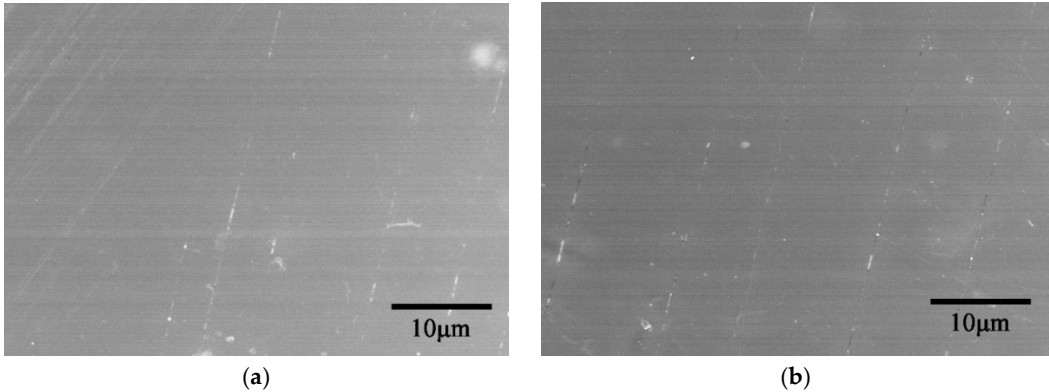

(**a**)                                                   (**b**)

**Figure 7.** SEM image of Kapton/Al$_2$O$_3$ surface nanocomposite film before (**a**) and after (**b**) curling.

### 3.3. Infrared Spectroscopy Analysis

Figure 8 shows the ATR-FTIR spectrum of the Kapton film and the Kapton/Al$_2$O$_3$ surface nanocomposite film. From curve a in the figure [35,36], two absorption bands appear in the range of 1800–1680 cm$^{-1}$, i.e., the 1774 and 1704 cm$^{-1}$ absorption bands that correspond to the imide I band and the II band, respectively. These refer to the symmetrical and asymmetric stretching vibrations of the two carbonyl groups (C=O) on a five-membered imine ring. The 1366 and 719 cm$^{-1}$ absorption bands correspond to the imide III band and the IV band, respectively, and are typical peaks of the Kapton film. The 1596 and 1494 cm$^{-1}$ absorption bands are stretching vibrations of the benzene ring in the carbon skeleton in the aryl ether. The 1232 cm$^{-1}$ absorption band is a stretching vibration of =C–O–C= in the

aryl ether and exhibits a broad absorption. The 1163 and 1111 cm$^{-1}$ absorption bands are attributable to in-plane angular vibration of =C–H on the substituted benzene ring. The 1012 cm$^{-1}$ absorption band is the in-plane rocking vibration of the para-substituted phenyl hydrogen atom. The absorption bands at 880 and 813 cm$^{-1}$ are the out-of-plane deformation vibrations of the isolated hydrogen atom and the ortho hydrogen atom of the benzene ring, respectively. The 633 cm$^{-1}$ absorption band is the bending vibration of the aromatic ether. These characteristic peaks are consistent with the Kapton film. From curve b, it can be seen that the infrared absorption peak position and the peak shape of the modified Kapton film were generally unchanged. After the polyimide film is treated by surface chemical modification via the ion exchange method, the nano-Al$_2$O$_3$ is precipitated in situ on the surface of Kapton film, and the depth of the infrared beam of ATR-FTIR on the sample is generally ~0.5–2 µm [37], which indicates that surface chemical modification has no effect on the internal structure of the Kapton film matrix.

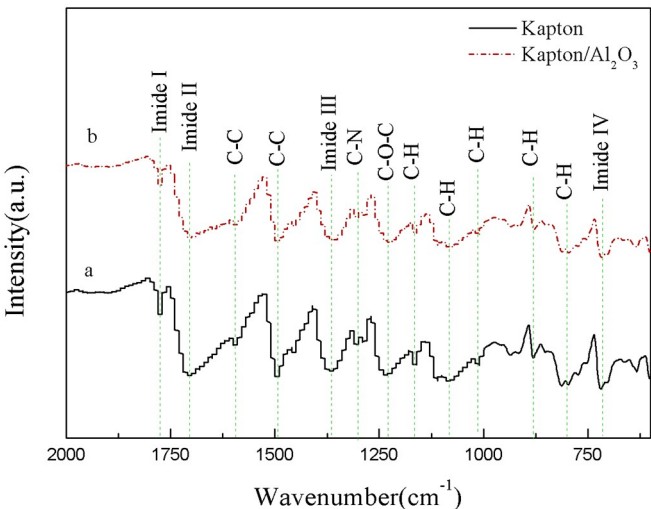

**Figure 8.** ATR-FTIR spectrum of the Kapton film and the Kapton/Al$_2$O$_3$ surface nanocomposite film.

## 4. Atomic Oxygen Environment Adaptability Evaluation

### 4.1. Morphology Analysis

Figure 9 shows the surface morphologies of the Kapton film and Kapton/Al$_2$O$_3$ surface nanocomposite film before and after atomic oxygen exposure. It can be seen that after exposure to atomic oxygen for 30 h, the surface of the Kapton film and the Kapton/Al$_2$O$_3$ composite film loses its luster, becomes rough, and presents a carpet-like morphology, as imaged by scanning electron microscopy [38]. Further comparison shows that the surface etch pit area (some examples marked with circles) of the Kapton film is larger after the atomic oxygen corrosion, while the surface etch pit area of the Kapton/Al$_2$O$_3$ composite film is relatively small, indicating that the Al$_2$O$_3$ surface inhibits the atomic oxygen erosion process to a certain extent.

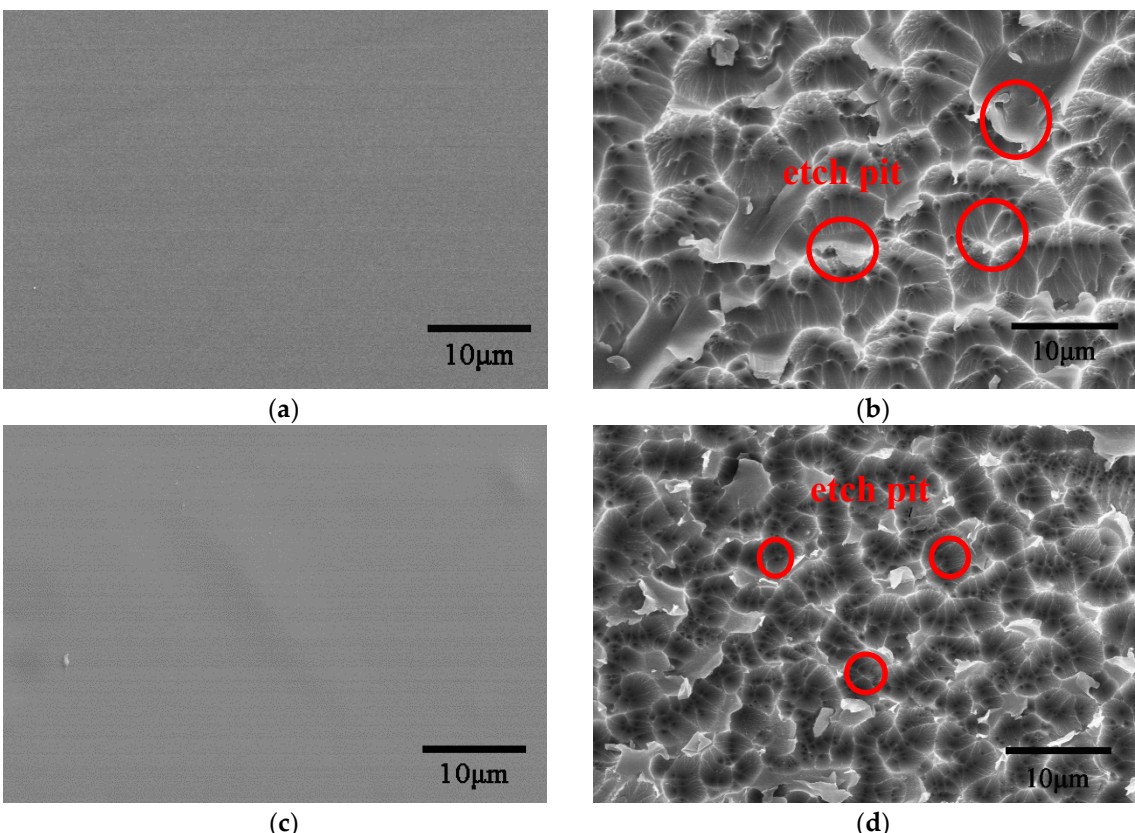

**Figure 9.** SEM image of Kapton film (**a**,**b**) and Kapton/Al$_2$O$_3$ (**c**,**d**) surface nanocomposite film before and after atomic oxygen exposure.

*4.2. Atomic Oxygen Weight Loss Analysis*

After atomic oxygen exposure, the Kapton film and Kapton/Al$_2$O$_3$ nanocomposite film showed mass loss, as shown in Figure 10. The figure shows the mass loss per unit area of the sample, and the dotted line is the fitted curve. Figure 10 demonstrates that, with increasing the atomic oxygen, the mass loss per unit area of the sample gradually increases. The mass loss curve of the Kapton film is linear, with a slope of about −1.2 and has a relatively large weight loss.

The oxidation weight loss process of Kapton/Al$_2$O$_3$ surface nanocomposite film can be divided into two stages: the pre-nonlinear stage and late linear stage. According to the change trend of the weight loss curve, the weight loss rate in the early stage is low and can be fitted as a parabola, as shown by the dotted line a in the first half of Figure 10. It is known from oxidation kinetics [39,40] that the Al$_2$O$_3$ layer on the surface of Kapton can separate the Kapton film matrix from the atomic oxygen molecules, preventing further diffusion of atomic oxygen to some extent and thus protecting the internal Kapton film matrix. In particular, the exposure time of the atomic oxygen ground experiment is less than 12 h. Due to the excellent anti-oxidation effect of Al$_2$O$_3$, the Kapton/Al$_2$O$_3$ surface nanocomposite film exhibits good antigenic oxygen performance. This is equivalent to a cumulative atomic oxygen flux of 1770 h at a distance of 400 km in space [41].

With increasing atomic oxygen erosion, the oxidation weight loss rate of the Kapton/Al$_2$O$_3$ surface nanocomposite film increases. When the exposure time of the atomic oxygen experiment reaches 18 h, the weight loss curve is basically parallel with the weight loss curve of the Kapton film, showing a linear relationship with a slope of approximately −1.2. This indicates that there is continued erosion by atomic oxygen and that the Al$_2$O$_3$ content in the nanocomposite film is gradually decreased. Then, holes gradually appear and the Kapton matrix is eventually exposed.

The atomic oxygen reaction coefficient of the Kapton film is generally $3.0 \times 10^{-24}$ cm$^3$/atom [10]. The average reaction coefficient of the Kapton/Al$_2$O$_3$ nanocomposite film at each stage in the atomic

oxygen exposure process is shown in Figure 11. It can be seen from the figure that the average atomic oxygen reaction coefficient of the Kapton/Al$_2$O$_3$ surface nanocomposite film is ~1.0 × 10$^{-24}$ cm$^3$/atom within 0–6 h, which is significantly lower than the Kapton film with a reaction coefficient of 3.0 × 10$^{-24}$ cm$^3$/atom. This indicates that there is improved resistance to atomic oxygen erosion in the modified Kapton film. With increasing atomic oxygen exposure time, the average atomic oxygen reaction coefficient in each stage of the composite film gradually is increased. When the atomic oxygen exposure time reaches 30 h, the atomic oxygen reaction coefficient is stable at 3.0 × 10$^{-24}$ cm$^3$/atom, which is consistent with the Kapton film reaction coefficient.

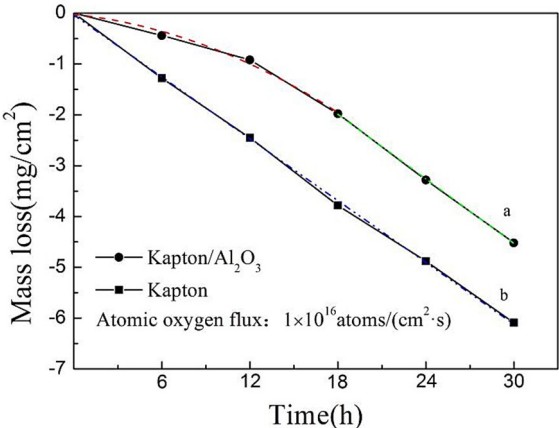

**Figure 10.** Mass loss after atomic oxygen exposure of the Kapton film and Kapton/Al$_2$O$_3$ surface nanocomposite film.

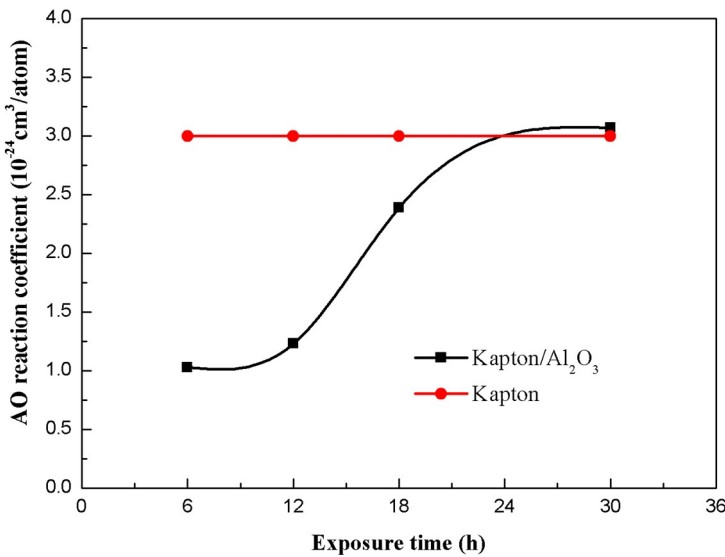

**Figure 11.** Atomic oxygen reaction coefficient of the Kapton/Al$_2$O$_3$ surface nanocomposite film.

### 4.3. X-ray Photoelectron Spectroscopy (XPS) Analysis

To further analyze the changes in the surface elements of the sample before and after atomic oxygen exposure and the atomic oxygen reaction mechanism of the film, the surface molecular structure and valence bonds of the sample before and after atomic oxygen exposure of the Kapton and Kapton/Al$_2$O$_3$ surface nanocomposite films were analyzed via XPS [42–45]. The XPS full scan spectrum is shown in Figure 12, and the surface element composition of the sample is shown in Table 1. After exposure to atomic oxygen for 30 h, the relative carbon content on the surface of Kapton and the Kapton/Al$_2$O$_3$ surface nanocomposite films decreased, while the relative content of nitrogen and oxygen increased.

The Kapton/Al$_2$O$_3$ surface nanocomposite film contains elemental Al in addition to the polyimide matrix elements C, O and N. After exposure to atomic oxygen, the relative aluminum content in the nanocomposite film reduced.

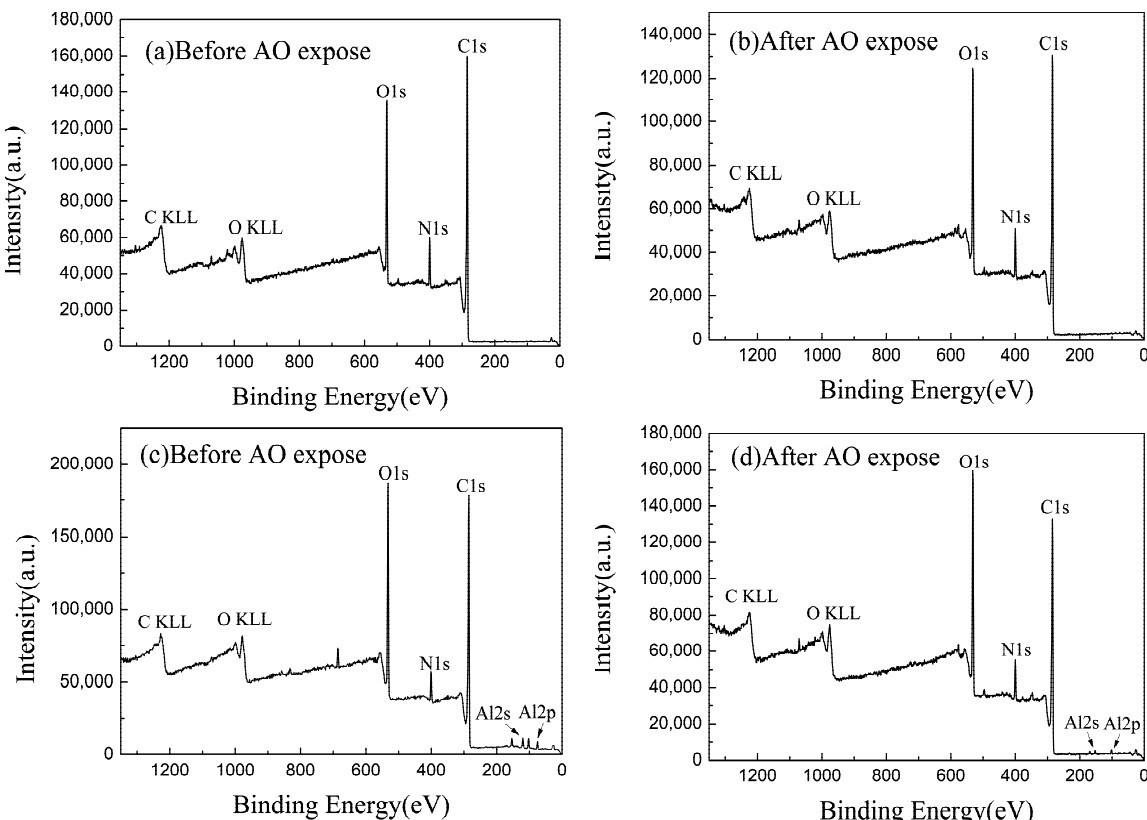

**Figure 12.** XPS full scan before and after atomic oxygen exposure: (**a**,**b**): Kapton, (**c**,**d**): Kapton/Al$_2$O$_3$.

**Table 1.** Relative content of elements before and after atomic oxygen exposure (atomic percentage).

| Materials | Condition | C1*s* | N1*s* | O1*s* | Al2*p* |
|---|---|---|---|---|---|
| Kapton | Before exposure | 74.85 | 6.96 | 18.19 | – |
| | After exposure | 71.98 | 7.2 | 20.83 | – |
| Kapton/Al$_2$O$_3$ | Before exposure | 65.6 | 6.96 | 23.3 | 4.13 |
| | After exposure | 61.63 | 10.87 | 26.45 | 1.05 |

### 4.3.1. Kapton Film

It can be seen from Table 1 that the relative atomic percentage of C in the Kapton film samples decreased by 2.87% after atomic oxygen exposure, and the relative atomic percentages of N and O elements increased by 0.24% and 2.64%, respectively. The decrease of C content can be attributed to the strong oxidizability of the atomic oxygen, which will chemically react with polyimide and be oxidized to CO$_2$ or CO, resulting in weight loss by oxidation of Kapton [9,10].

Figure 13 shows the high-resolution spectra of the C1*s* and O1*s* regions and the peak fitting curves using the XPSPEAK software (version 4.1) before and after atomic oxygen exposure of Kapton films. There are four different states of carbon atoms in the C1*s* spectrum, with the fitted peaks are located at binding energies of 284.37 eV (corresponding to the C–C bond in the ODA benzene ring), 285.18 eV (representing the C=C bond in the PMDA benzene ring), 285.90 eV (C–O bond), and 288.41 eV (C=O bond or C–N bond), respectively. After exposure to atomic oxygen, the above fitted peaks still exist, but the relative area of the peak at the lower binding energy of 284.37 eV is reduced, causing an overall movement to lower binding energies of 284.29, 285.01, 285.81, and 288.39 eV, respectively. This indicates

that the C–C bond in the ODA benzene is broken after atomic oxygen exposure, and the carbon is oxidized to form a volatile oxide, such as CO and $CO_2$, which eventually leads to decomposition of the imine ring.

On further analyzing the chemical valence bond of oxygen, the spectrum before atomic oxygen exposure can be fitted to two peaks at 531.69 and 533.03 eV, corresponding to the C=O bond (corresponding to oxygen of carbonyl group in polyimide molecule) and C–O bond (corresponding to oxygen bridge of diphenyl ether in polyimide molecule), respectively. After exposure to atomic oxygen, these two peaks slightly shifted toward a higher binding energy of 531.70 and 533.07 eV, respectively. It is worth noting that the relative peak area and peak intensity of the C–O bond in ODA decreased. To some extent, oxygen bridge of diphenyl ether is related to the flexibility of Kapton. This suggests that the C–O structure may be more sensitive to atomic oxygen exposure, and the atomic oxygen participates in the reaction to form a more stable C=O structure, which weakens the flexibility of Kapton molecular chain by affecting the C–O bond in ODA and is consistent with the analysis of the C1*s* XPS spectrum.

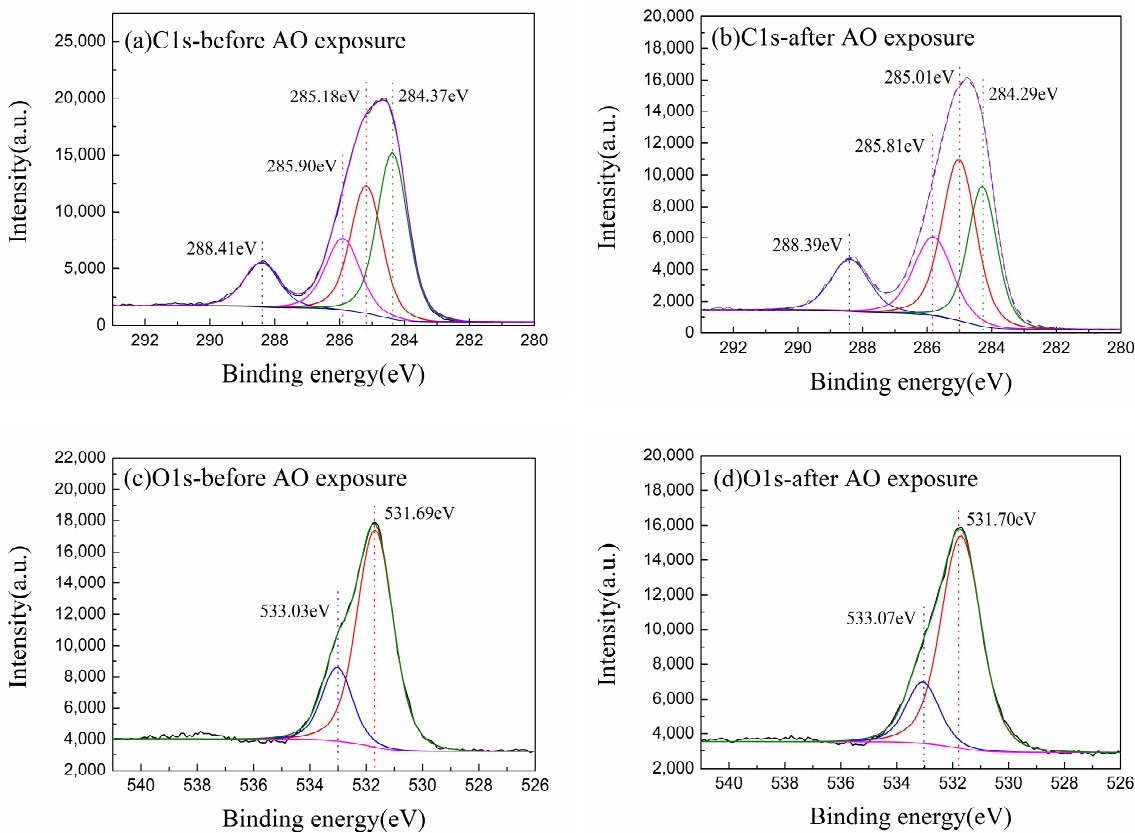

**Figure 13.** High-resolution spectra of C1*s* and O1*s* before and after exposure of the Kapton films: (**a**,**b**): C1*s*; (**c**,**d**): O1*s*.

### 4.3.2. Kapton/$Al_2O_3$ Composite Film

The procedure for preparing Kapton/$Al_2O_3$ nanocomposite film is outlined in Figure 2. The Kapton film was subjected to a hydrolysis reaction in an aqueous alkaline (KOH) solution to open the polyimide and form a potassium polyamic acid salt, which then reacts with an aqueous solution of $AlCl_3$ to form an aluminum polyamic acid salt via ion exchange. Finally, the aluminum polyamic acid salt is heat-treated under oxygen-containing conditions to imidize the polyamic acid salt, release water molecules and regenerate the polyimide structure. During this process, the $Al^{3+}$ moves out from the polyamide acid and becomes an Al atom, which reacts with oxygen and forms $Al_2O_3$ molecules or submicroscopic particulates. By prolonging the heat treatment time or increasing the heat treatment

temperature, the clusters of particles can be stacked to form larger nanoparticles, which distribute uniformly on the modified layer and aggregating on the surface of the polyimide. The cross-section SEM observation and EDS line scanning are shown in Figure 14. An aluminum-containing layer with thickness of about 10 microns is formed on the surface of Kapton.

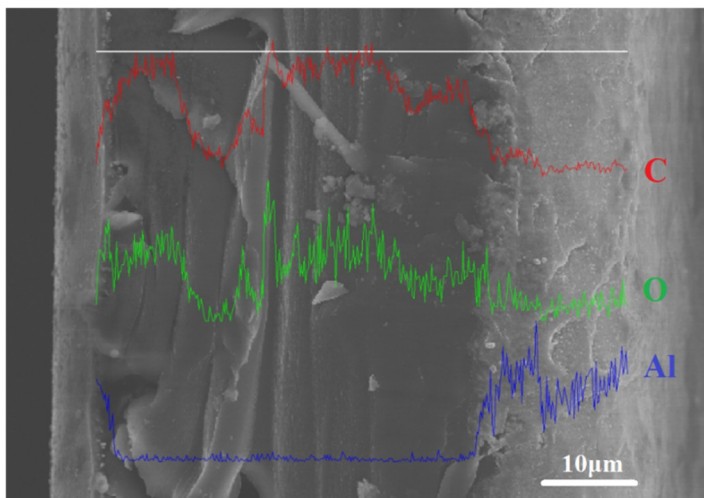

**Figure 14.** EDS line scanning image for cross-section of composite films.

The high-resolution XPS spectrum of the Al2*p* and O1*s* regions of the Kapton/Al$_2$O$_3$ composite films is shown in Figure 15. The Al2*p* peak is located at 74.83 eV, which is consistent with the Al2*p* peak in Al$_2$O$_3$ in the literature [46,47]. The O1*s* region XPS spectrum on the surface of the composite film was subjected to peak deconvolution. The peak positions at binding energies of 533.16 and 531.94 eV correspond to the C–O and C=O bonds in the polyimide structure, respectively. The peaks at 531.25 eV are consistent with the Al–O bond binding energy in Al$_2$O$_3$. The above results indicate the successful preparation of the Kapton/Al$_2$O$_3$ nanocomposite film via the surface-modification–ion-exchange method.

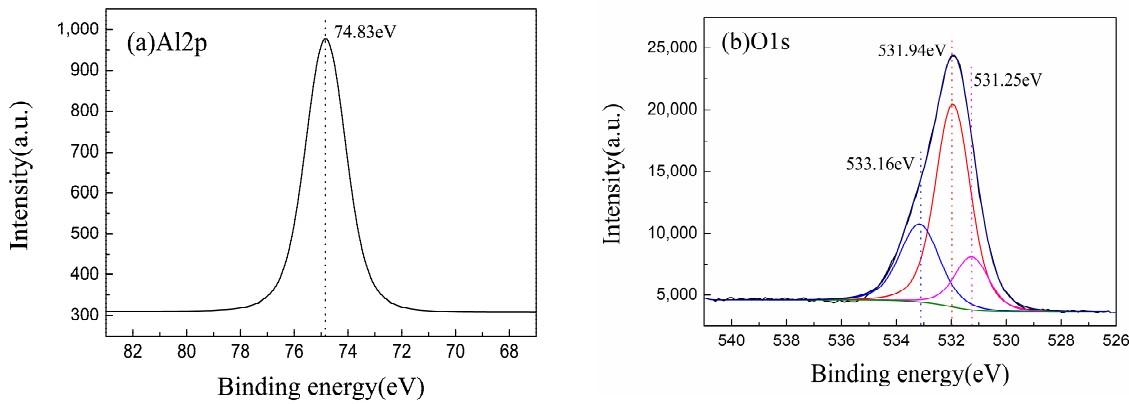

**Figure 15.** XPS spectrum of Kapton/Al$_2$O$_3$ surface nanocomposite film: (**a**) Al2*p*. high-resolution spectrum; (**b**) O1*s* high-resolution spectrum.

Figure 16 shows the high-resolution spectra of the C1*s* and O1*s* regions and the peak fitting curves using XPSPEAK software before and after exposure of the Kapton/Al$_2$O$_3$ nanocomposite films to atomic oxygen. There are also four different states of carbon atoms in the C1*s* spectrum, with binding energies of 284.31, 284.94, 285.82, and 288.46 eV. After exposure to atomic oxygen, the relative area of the peak at a binding energy of 285.82 eV was significantly reduced, while the relative area of the peak located at 288.46 eV significantly increased. Finally, the binding peaks of the fitted peaks in the C1*s*

spectrum changed to 284.29, 285.22, 286.27, and 288.13 eV. This indicated that for the Kapton/Al$_2$O$_3$ nanocomposite film, atomic oxygen mainly damages the C=C bond in the PMDA benzene ring and has an enhanced effect on the C=O bond.

The Kapton/Al$_2$O$_3$ composite film O1$s$ high-resolution spectral fitting peaks located at binding energies of 531.25, 531.94, and 533.16 eV, correspond to Al–O bonds, C=O bonds, and C–O bonds, respectively. After atomic oxygen exposure, the relative area of the peak at 531.25 eV significantly reduced. The binding energy of the fitted peaks in the O1$s$ spectrum were reduced to 530.95, 531.68, and 532.94 eV, respectively. In addition, the results in Table 1 show that the elemental content of Al in the surface layer of Kapton/Al$_2$O$_3$ composite film decreased from 4.13% to 1.05%. These results indicate that the Al$_2$O$_3$ surface damage after 30 h of atomic oxygen exposure cannot effectively protect the Kapton matrix. This is mainly due to the fact that, for the Kapton/Al$_2$O$_3$ composite film prepared by the ion exchange method, the Al$_2$O$_3$ surface layer that was formed by the deposition of Al$_2$O$_3$ molecules or fine particles is not dense. For a short atomic oxygen exposure period, the Al$_2$O$_3$ surface layer can impede atomic oxygen erosion and diffusion. However, on prolongation of the atomic oxygen exposure time, atomic oxygen diffuses into the interior of the Kapton matrix through the pores of the Al$_2$O$_3$ layer, causing damage to the substrate and eventual erosion of the Al$_2$O$_3$ surface layer.

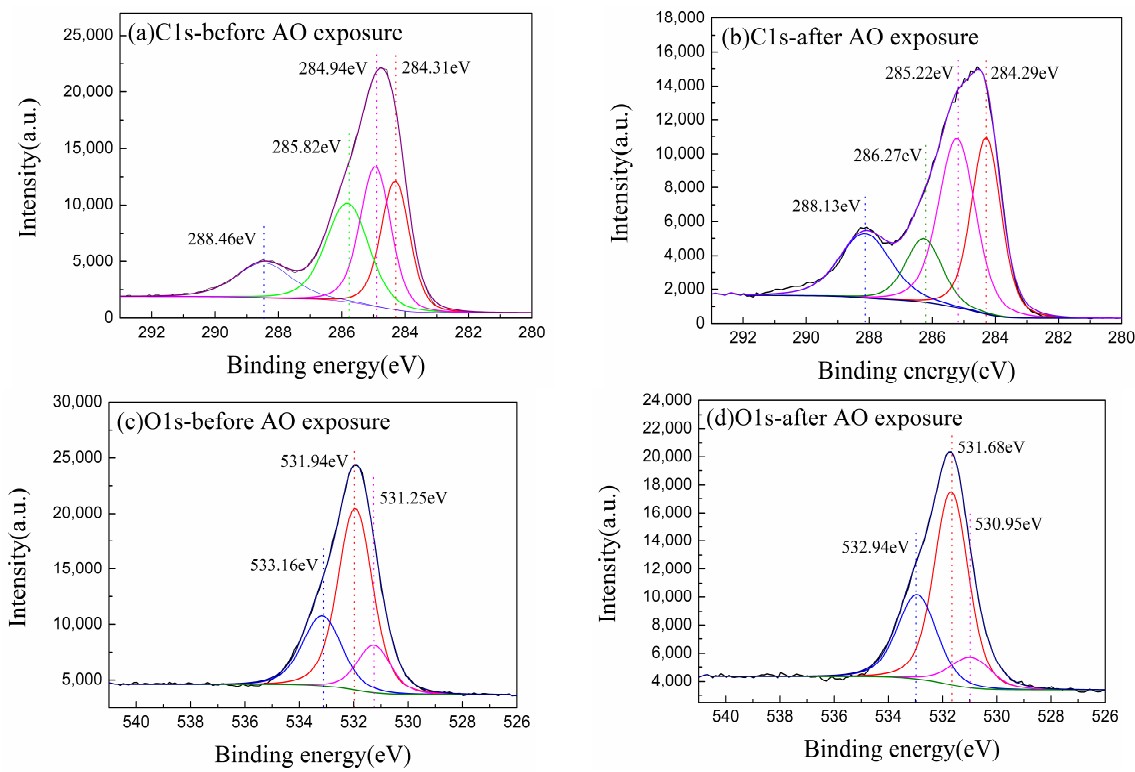

**Figure 16.** High-resolution spectra of C1$s$ and O1$s$ before and after exposure to Kapton/Al$_2$O$_3$ films: (**a**,**b**): C1$s$; (**c**,**d**): O1$s$.

## 5. Conclusions

- Compared with the Kapton film, the Kapton/Al$_2$O$_3$ surface nanocomposite film prepared via the ion exchange method had no effect on the internal structure of the Kapton film matrix. However, the optical transmittance decreased by ~10% in the wavelength range of 500–800 nm for the enhancement of diffuse reflection caused by Al$_2$O$_3$ particles on the surface. The tensile strength and elongation were much higher than for the pure Kapton film and demonstrated its good flexibility, which is due to hindering initiation and propagation of surface cracks by Al$_2$O$_3$ particles in the subsurface layer for the existence of phase interface.

- Under the action of atomic oxygen erosion, the surface of the Kapton film and the Kapton/$Al_2O_3$ composite film showed a carpet-like morphology, and the surface of Kapton/ $Al_2O_3$ composite film had relatively small corrosion pits. In contrast to C–C bond rupture in ODA benzene of the Kapton film, the C=C bond in the PMDA benzene ring was mainly destroyed in the Kapton/$Al_2O_3$ nanocomposite film; finally, the carbon can be oxidized to form volatile small molecular substances, such as $CO_2$ or CO.

- At short exposure times to the atomic oxygen environment, the $Al_2O_3$ layer on the surface of Kapton can inhibit the erosion and diffusion of atomic oxygen. The corrosion weight loss rate and the atomic oxygen reaction were lower. The Kapton/$Al_2O_3$ surface nanocomposite film demonstrated an improved resistance to atomic oxygen erosion. With increasing exposure time to atomic oxygen, the $Al_2O_3$ layer prepared by ion exchange became less dense. The atomic oxygen diffused into the Kapton matrix via the pores of the $Al_2O_3$ layer, causing detachment from the substrate. This results in a loss of the protective surface $Al_2O_3$ layer and exposure of the Kapton matrix, and the atomic oxygen reaction coefficient is consistent with the Kapton matrix.

**Author Contributions:** Investigation, D.J. and D.W.; Writing—Original Draft Preparation, D.J.; Methodology, D.W.; Conceptualization, G.L.; Project Administration, Q.W.; Supervision, Q.W.; Writing—Review and Editing, Q.W.

**Funding:** This work is supported by the Natural Science Foundation of China (No. 51873146) and Natural Science Foundation of Hebei Province (No. E2019202106).

**Conflicts of Interest:** The authors declare no conflict of interest.

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
