# Peer review of "Atomic Oxygen Adaptability of Flexible Kapton/Al2O3 Composite Thin Films Prepared by Ion Exchange Method"

_coatings, doi:10.3390/coatings9100624_

Round 1
Reviewer 1 Report
The paper presents an interesting experimental study on the surface modification of Kapton films with Al2O3 in order to improve its atomic oxygen erosion in space. The topic is quite interesting. I recommend the publication after major revisions listed below.
Introduction page 2: The authors should more deeply analyze the state of the art on the ion exchange method used to functionalize the Kapton surface. Some works are cited but not discussed and other should be added. The originality of he work should more emphasized. Experimental: a scheme of the preparation procedure should improve the paper quality2 The choice of the processing parameters has not discussed at all in the Experimental section. Is it possible the study of the properties depending on the processing parameters (time of hydrolysis with KOH and time of the heat treatment)? Is it possible to quantify the Al2O3 content and its dependence of the processing parameters? Is it possible to quantify the nanocomposite thickness in the film? The Mechanical properties are only presented buy not discussed. Is the mechanical behavior expected? Why? Could the authors correlate the mechanical performance with the Al2O3 content?
Reviewer 2 Report
Authors reported oxygen adaptability of kapton/Al2O3 hybrid films obtained via ion exchange methos. This work is interesting. However, authors need to improve and address the following concerns: 1. In abstract “XPS analysis showed that the etching pits had a carpet-like morphology on the 24 composite film surface and were relatively small after atomic oxygen erosion” please correct this sentence.. it is not XPS ..It is SEM analysis. 2. In XPS, O1s shows the oxygen deficiency peaks at 533 eV. More emphasis the role of this peak and extend the explanation accordingly. 3. How is overcome the brittleness of this hybrid? 4. Is it optimal concentration Kapton? 5. Importance of Al2O3 should be highlighted in introduction and add this relevant article: Materials Research Bulletin 96, 233-245 6. Thickness variation of film vs stress/strain should be analyzed more in details.
Reviewer 3 Report
This article is written well and can be an addition to the research of the scientific community in the area of polyimide films. However, there are a few issues that the author needs to resolve. I would recommend publication of this paper once all the issues in the comments are addressed adequately.
Fig3- mention a and b in the caption text. I am assuming that nanocomposite film is b.
Figure 3, 6 and 8- scalebars either missing or not visible. Author should make the scalebars and scale measurement clear and visible.
How many times the stress-strain experiment of Kapton and composite films was carried out? One experiment to calculate and observe the elongation at break is not sufficient. The norm is, at least 3 experiments.
FTIR suggest good and important results although Figure 7 is not easily comprehensible. Nowadays, the authors highlight important vibrations related to this IR within the figure, making it easier to compare between the spectra. Following are examples of references for highlighting ftir peaks:
-Carbon, 98 (2016), pp. 491-495, DOI: https://doi.org/10.1016/j.carbon.2015.10.083
-Nanoscale Research Letters, 2013, 8(1):32, DOI: https://doi.org/10.1186/1556-276X-8-32
For Morphology analysis, what is the surface etch pit (line 185)? In fig 8, the grooves are big and within big grooves, there are smaller grooves.
Does the author refer to smaller or the larger imperfections in the surface for calculations and comparison of pit area size?
Could the author mention the differences of this work from "Kapton/Al2O3 surface nanometer composite membrane and application. China Patent# CN102418096A" It would be interesting and desirable for the perspective readers.
Lines 328 to 330. The two sentences are contradictory. Transmittance values changed after ion exchange experiment. So, there must be a change in the structure too, that's how optics works. Author should rephrase the sentences or elaborate more on the internal structural parameters that are unaffected while which factors are affected. Author must make sure that the %Transmittance is due to surface modification only.
Round 2
Reviewer 1 Report
The authors have addressed most of my comments.
The paper is now ready for publication.
Reviewer 2 Report
Authors improved the manuscript and i recommended for acceptance
Reviewer 3 Report
Changes including SEM picture are satisfactory.